# Autonomous Face Classification Online Self-Training System Using Pretrained ResNet50 and Multinomial Naïve Bayes

**DOI:** 10.3390/s23125554

**Published:** 2023-06-14

**Authors:** Łukasz Maciura, Tomasz Cieplak, Damian Pliszczuk, Michał Maj, Tomasz Rymarczyk

**Affiliations:** 1Research and Development Center, Netrix S.A., 20-704 Lublin, Poland; lukasz.maciura@netrix.com.pl (Ł.M.); damian.pliszczuk@netrix.com.pl (D.P.); michal.maj@netrix.com.pl (M.M.); tomasz.rymarczyk@netrix.com.pl (T.R.); 2Department of Organization of Enterprise, Faculty of Management, Lublin University of Technology, 20-618 Lublin, Poland; 3Faculty of Computer Science, WSEI University, 20-209 Lublin, Poland

**Keywords:** face recognition, autonomous systems, online learning, Multinomial Naïve Bayes classifier, convolutional neural networks

## Abstract

This paper presents a novel, autonomous learning system working in real-time for face recognition. Multiple convolutional neural networks for face recognition tasks are available; however, these networks need training data and a relatively long training process as the training speed depends on hardware characteristics. Pretrained convolutional neural networks could be useful for encoding face images (after classifier layers are removed). This system uses a pretrained ResNet50 model to encode face images from a camera and the Multinomial Naïve Bayes for autonomous training in the real-time classification of persons. Faces of several persons visible in a camera are tracked using special cognitive tracking agents who deal with machine learning models. After a face in a new position of the frame appears (in a place where there was no face in the previous frames), the system checks if it is novel or not using a novelty detection algorithm based on an SVM classifier; if it is unknown, the system automatically starts training. As a result of the conducted experiments, one can conclude that good conditions provide assurance that the system can learn the faces of a new person who appears in the frame correctly. Based on our research, we can conclude that the critical element of this system working is the novelty detection algorithm. If false novelty detection works, the system can assign two or more different identities or classify a new person into one of the existing groups.

## 1. Introduction

Autonomous intelligent systems, e.g., robots or intelligent multimodal chatbots (with vision and sound sensors), should have the possibility of the unsupervised learning of new, unknown objects for further recognition. In this paper, we focus on recognizing unknown earlier persons without the significance of the persons’ names. An intelligent robot should be able to learn similarly to a small child or animal that does not know a language, however can recognize a person in the environment. Other attributes of acquainted persons such as their name, timbre, relationship, etc., could be later associated with a given person identifier in the autonomous system’s memory.

The memory model of a self-learning system should be designed to create such an autonomous intelligent system. One of the challenges of creating a system is that most machine learning algorithms, such as artificial neural networks, cannot update a model using new classes because all classes should be known during the training process. Moreover, the autonomous system should update a model in real-time without a long training process. However, pretrained convolutional neural networks could be used for encoding data such as face images.

The general-purpose online training systems are created for learning sequences. Paper [1] presents Hierarchical Temporal Memory, which can remember a time series encoded to distributed representations. In paper [2], the self-learning memory can remember, classify and reproduce symbolic sequences whereas in paper [3] the memory can remember and classify time series sequences.

There are some unsupervised recognition systems available. Paper [4] describes the unsupervised system, without using any labeled data. The system used a StyleGAN network, pretrained without labeled data to train a prototypical network that can identify human faces. Paper [5] uses GAN-based augmentation, which learns to maximize the similarity between two augmented images of the same synthetic instance, leading to high recognition accuracy. The unsupervised autonomous learning system, which learns from a video sequence, is presented in paper [6]. First, images from video sequences are extracted. Then, after an extraction of features using the Gradient Boost Algorithm, they are matched with features of images belonging to templates collected earlier from known classes. Finally, the system uses SVMs for the actual authentication process.

In the autonomous intelligent system, novelty detection is very important to recognize if the analyzed person is new or previously known. If the analyzed person is new, the system should remember it and update the model using its samples. Otherwise, the system should recognize it (using previously trained classes). There are multiple novelty detection algorithms available. Some are based on autoencoders [7] for outlier detection (novel data will be outliers). Unfortunately, while using autoencoders for outlier detection, the previously pretrained autoencoder on known data is needed, which is impossible when the known dataset will be changed during system operation.

In recent years, Generative Adversarial Networks models have been used for novelty detection [8]. GANs consist of a generator that generates new artificial data and a discriminator which is learning to detect real and artificial data. Training both models together improves them and constitutes a zero-sum game. Finally, the discriminator is used as a novelty detection model. Unfortunately, this method could not be used in an autonomous learning system because it additionally needs a set of known data before training, and it will be changed during work, and training GANs each time in real-time is impossible.

Other systems with a semi-supervised faces recognition ability are available in the literature. Paper [9] presents the method for estimating facial attractiveness in images and videos. The proposed semi-supervised algorithm combines multiple graphs to find a unified flexible manifold embedding model. In paper [10], the evaluated Kullback–Leibler divergence was used for gender classification in images of human faces. Paper [11] presents an algorithm for face matching and gender prediction. The face identification process uses distance metrics. In paper [12], feature extraction techniques and a k-NN statistical classifier were used to solve face recognition problems.

Article [13] is a survey of class incremental learning algorithms for visual tasks. The problems related to the ability of artificial agents to increment their capabilities while confronting new data and multiple artificial intelligence solutions were presented.

Paper [14] presents a system that is a similar solution to what is presented in this paper, based on the Convolutional Neural Network for image encoding and an incremental SVM algorithm as a classification model. This system uses a new sample to incrementally update the model; however, there is no possibility of training new classes in the created system.

Except for this paper, there are some online machine learning systems available on the GitHub platform. The online machine learning library in Python called River [15] enables training a model that can learn from new data without having to revisit past data, which is robust to concept drift. The Vowpal Wabbit [16] is a machine learning system that pushes the frontier of machine learning, with methods such as online, hashing, all-reduce, reductions, Learning2Search and active and interactive learning.

The main difference between the created system shown in this paper and the other described systems is that this paper’s system can automatically learn new persons from a camera or video file without any previously collected data about those persons. This property additionally refers to a new, novelty detection algorithm: a system element. The created system was tested on a video file downloaded from the Internet. The persons who appeared in the frame were used for the automatic training and inference of the system, resulting in all correct classifications, confirming that the system works properly.

The research methods are described in the second section of this paper. The third section describes the initial research that leads to choosing some of the methods that will be elements of the final system. The fourth section describes the final version of the created system with its main elements: cognitive tracking agents and a novelty detection algorithm. Finally, the experiments using the created system on a video file with multiple persons appearing several times are described in the fifth section.

The main contribution of the described research is the cognitive tracking agent that can track a person’s face and autonomously train the model (even for new, non–existing persons in the model), based on the incoming video data. The whole system consists of multiple such agents which can independently serve several people that appear in the incoming frames from a camera or video file.

## 2. Description of Used Methods

### 2.1. ResNet50 Model

The ResNet50 network is a convolutional neural network with 50 layers, trained using a special deep residual learning algorithm [17]. Deep residual learning uses groups of stacked layers in the form of blocks, which can be defined as follows:$$y=F\left(x,\left\{W_{i}\right\}\right)+x$$
where
*x* is the input vector;*y* is the output vector;$F\left(x,\left\{W_{i}\right\}\right)$ represents the residual mapping to be learned.

For instance, in Figure 1 there is a block with 2 layers which could be described in the following formula:$$F=W_{2}\sigma \left(W_{1}x\right)$$
where
*σ*—denotes ReLU without biases (for simplification).

**Figure 1 sensors-23-05554-f001:**
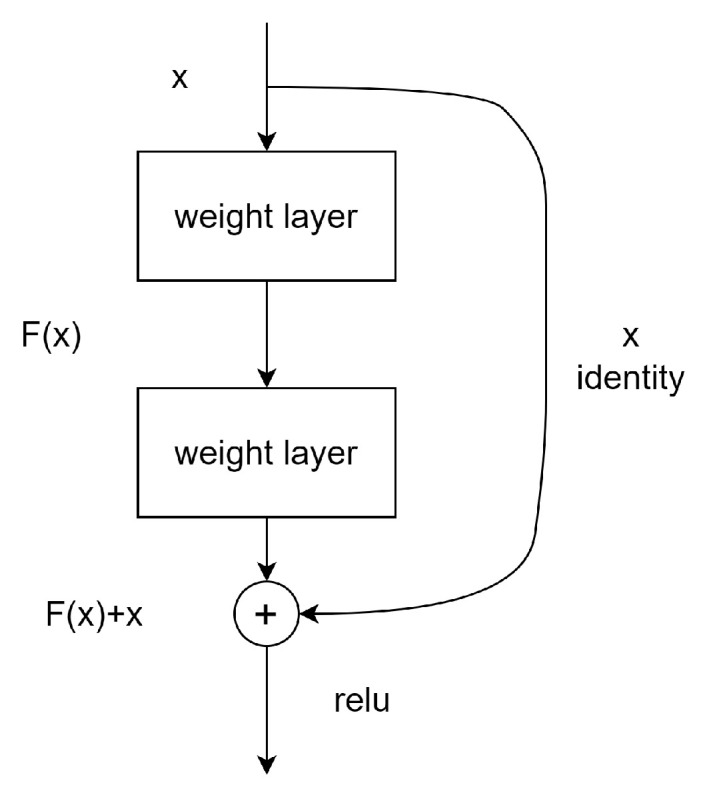
Sample of a building block in residual learning.

In this solution, the addition operation of *F* and *x* is computed by a shortcut connection with element-wise addition. This connection does not introduce any additional parameter and does not influence computation complexity.

Using residual networks, convolutional neural networks allow the training of networks containing multiple layers more efficiently than classical convolutional neural networks. The architecture of ResNet50 used in this research contains 16 residual blocks and a classification head. Each residual block contains a convolutional layer 1 × 1, batch normalization layer, activation, convolutional layer 3 × 3, batch normalization layer, activation, convolutional layer 1 × 1, batch normalization layer and activation.

### 2.2. Naïve Bayes Classifier

Naïve Bayes Classifier is a group of methods [18,19,20,21] for supervised classification using Bayes’ theorem, including Multinomial Naïve Bayes [18,19], Bernoulli Naïve Bayes [18,19] and Gaussian Naïve Bayes [20]. The Bayes’ theorem is used as follows:(1)$$P\left(y|x_{1},\ldots ,x_{n}\right)\propto P\left(y\right)\prod_{i=1}^{n}P\left(x_{i}|y\right)$$
and as a result, the classification rule can be defined by the following equation:(2)$$\hat{y}=\arg max\left(P\left(y\right)\right)\prod_{i=1}^{n}P\left(x_{i}|y\right)$$

The multinomial distributed data are used in the Multinomial Naïve Bayes classification algorithm, [18,19] version. The next step is the vector count parametrization for each class y:(3)$$\theta_{y}=\left(\theta_{y1},\ldots ,\theta_{yn}\right)$$
where *n* is the number of discretized values, and $\theta_{yi}$ is the probability $P\left(x_{i}|y\right)$ of discretized value *i* in the sample from class *y*. Next, the parameters $\theta_{y}$ are estimated using smooth maximum likelihood (smooth version), namely employing relative frequency counting:(4)$${\hat{\theta }}_{yi}=\frac{N_{yi}+\alpha }{N_{y}+\alpha n}$$
where $N_{yi}$ is the number of times that discretized value *i* appears in a sample of class *y* in the training set *T* which could be defined by the following:(5)$$N_{yi}=\sum_{x\in T}x_{i}$$
whereas a total count of all discretized values for class *y* is defined by the following:(6)$$N_{y}=\sum_{i=1}^{n}N_{yi}$$

The priors for smoothing α≥0 account for discretized values not existing in the learning samples and avoid zero probabilities in further calculations.

### 2.3. Support Vector Machines

Support vector machines [22,23] are classifiers, which find an optimal hyperplane to separate classes in a high-dimensional space based on given examples (as we can see in Figure 2). An optimal hyperplane is designated after finding maximized margins based on given samples.

In the SVMs, different kernels could be used such as linear, RBF, sigmoid and others. In the current research, the RBF kernel was used which could be defined as follows:$$K\left(x,x^{\prime }\right)=e^{-\frac{{\left\|x-x^{\prime }\right\|}^{2}}{2\sigma^{2}}}$$
where ${\left\|x-x^{\prime }\right\|}^{2}$ is the squared Euclidean distance between the 2 feature vectors, and σ is a feature vector.

## 3. Initial Experiments

### 3.1. Detection and Encoding of Input Faces

The images of faces seized from the camera are extracted using the real-time *Viola-Jones* algorithm [24] from the *OpenCV* library. Following this, the face images are scaled into the proper shape (224 × 224 × 3) and encoded using the *ResNet50* convolutional neural network [17], pretrained on the *VGGFace2* dataset [25] with a removed classifier (last dense layers). The used pretrained model was downloaded from [26]. Finally, the output of the network is produced by the last convolutional layer flattened to a vector with a size equal to 2048 elements.

### 3.2. Choosing the Proper Classification Model

The initial experiments were performed to choose the best classifier to enable a model update for new classes. The Pins Face Recognition dataset from Kaggle containing 105 classes (downloaded from [27] and reduced to images with good resolution) was encoded after proper face image extraction, as was described previously (only full-face images were taken into account). Finally, 5287 samples were obtained from the training dataset and 1763 samples from the test dataset.

For the dataset described above, the classifiers based on Naïve Bayes [18,19,20,21] and other different classical classifiers [22,23,28,29,30] were tested (some of them have the possibility of updating a model by a new class). The results of each classifier are presented in Table 1.

As we can see, the best results were obtained using Multinomial Naïve Bayes. Moreover, this model can be updated using a novel class. The only limitation of the updated model is that the maximum number of available class labels should be established at the beginning of model creation. The complexity of the Multinomial Naïve Bayes model depends on this maximal number of available classes and that it does not change while updating a model with new classes or samples from existing classes, because in the model the statistics for each class are kept from the beginning.

## 4. Description of the Created System

The created system processes each image in a video sequence (from a camera or video file). In each input image, all face images are detected and extracted. After extraction, the faces are assigned to existing cognitive tracking agents (with similar positions), or a new cognitive tracking agent is created in the system. If a given face does not appear in the following images in the sequence in a similar position, then the given agent is removed from the system. All cognitive tracking agents operate with a face recognition model. The extracted faces, using the 468 landmarks detection algorithm [31] from the MediaPipe library are encoded using the pretrained convolutional model described in Section 2, and the vector representing a given face is moved to the proper tracking agent. The landmarks are used for the boundary of face extraction, which is converted into a quadratic shape using the vertical size (extension of the cut area on the horizontal axis). Moreover, the cut area is additionally extended by 10% of the previously designated area. Since the MediaPipe landmarks used have 3D positions, the directions of the faces are additionally computed. Only faces in horizontal directions with ranges of <−20, 20> degrees are considered in further processing.

Figure 3 presents the system’s core operation. Faces from 3 persons are detected along a subsequent frame, then face images and positions are sent to the core of the system which encodes these face images and allocates them to a proper agent which is tracking this person as was described earlier. Each active agent generates some outputs for each frame, where the face assigned to the agent exists. The outputs for each agent are the following: person ID (or if a face is not recognized yet after it appears), person name and current agent state in the form of text (for instance, “Novelty detection” or “Training person 105” etc.).

### 4.1. Cognitive Tracking Agents

The cognitive tracking agent is an object which tracks one face in the video sequence and deals with the machine-learning model. The system assigns each face in the current frame to one tracking agent, with a minimal distance between the position of the given face and the last position of the face saved by the agent. For example, if the calculated minimal position is greater than the distance threshold, then in that case, the face is not assigned to any tracking agents, and a new tracking agent is created and added to the current tracking agent list. If, after several frames are given tracking, the cognitive agent is not updated, then it means that the given face simply disappeared from the visual scene; as a result, the tracking agent is removed from the system. If the same face is further available in the scene, then it will be processed by the new cognitive tracking agent.

Besides the tracking function, each tracking agent contains a machine state that can deal with the face recognition model. The machine state consists of the following stages:Novelty detection stage: in this stage, the novelty detection algorithm is working and detects if a given face is novel (not existing in the set of classes trained earlier) over *N*_0_ = 10 samples. In the case of novelty, the agent switches to stage 1. Otherwise, the agent switches to stage 3;Initial training stage: in this stage, the vectors (encoded face images) collected since the tracking agent started working are used to train a new class. In this stage, a label for a new class is established and saved in the agent. After initial training using samples collected during the agent working, this agent switches to stage 2;Continuation of training stage: this stage uses new samples for training a given class, using the label established in the previous stage and new vectors. After training using *N*_1_ = 10 samples, the agent switches to stage 4 (because now the label of the face is known, the 3rd stage can be omitted);Initial recognition stage: the agent predicts the existing (not novel) class, using a dominant label calculated based on a set of collected samples (at least *N*_2_ = 5) in this stage. If most of the predictions are from the same label, then the label is saved as recognized, and the agent switches to stage 4;Analyzing stage: the agent is analyzing predictions of the known face, and when the predictions are different from the proper label, it updates the model using the wrongly recognized sample with the proper label;Removement stage: the agent switches to this stage from any other stage if the tracked face disappears and then after *N*_3_ = 10 frames without the face being detected close to tracking the last position. The agent in this stage is nonactive and will be removed by the system from the agents lists.

The values *N*_0_, *N*_1,_
*N*_2_ and *N*_3_ are established after some experiments during the creation of the system. A diagram of the flow between stages in the cognitive tracking agent machine state is presented in Figure 4.

### 4.2. Novelty Detection Algorithm

The novelty detection algorithm is a very important element of the created system. It is necessary to decide to predict a known class or a novel class training. The created novelty detection algorithm is based on Multinomial Naïve Bayes model responses for a new vector. Because the responses (except the dominant class response) have very small (even equal to zero) values, the logarithms of responses will be processed. There very often occurs situations where the second maximal value is zero, because the response value is so small that it does not fit in the float variable; therefore, the processing of logarithms responses is necessary (in the Scikit-learn library, the Multinomial Naïve Bayes response is additionally available in the form of logarithms).

In the novelty detection algorithm, the SVM model [23] was used to recognize novelty samples based on the response distribution of the Multinomial Naïve Bayes model. The training data were prepared in the following way for the novelty detection training process: The data with vectors from 105 celebrities were initially divided into 2 sets. The first set (53 celebrities) was used to train the Multinomial Naïve Bayes classifier, and the remaining data were to be used as novel data to test the responses of the classifier only. After training the classifier, the data from the classes used from training (53 classes) and the data from the remaining classes (52 classes) were used to obtain classifier responses in the form of logarithms. After that, all responses were sorted in non-declining order, and the 20 highest logarithms were saved for each tested sample. All samples with known data were labeled with a value of zero, whereas all samples with novel data were labeled with a value of 1 (which means that novelty was detected).

The above data, generated using the Multinomial Naïve Bayes model were used to train the SVM model for novelty detection. After testing the SVM model, the level of accuracy of novelty detection was 97.14% for training data and 96.48% for test data. This level of recognition was obtained using one frame only. Nevertheless, this novelty detection SVM model will be run on multiple frames in the created system.

In the final novelty detection algorithm implemented in the created system along with *N*_0_ (experimentally set to 10) samples, the novelty score value is calculated using an average of predictions from novelty detection:(7)$$\overset{-}{X}=\frac{\sum_{i=1}^{N_{0}}p_{i}}{N_{0}}$$
where *p_i_* is the prediction of novelty level from range <0, 1>, obtained based on a single sample.

This average score was analyzed using experiments on the real videos, and finally, a face analyzed along with *N_0_* samples is considered novel in the case where the following condition is met:(8)$$\overset{-}{X}>T_{n}$$
where *T_n_* is the novelty score threshold set to 0.85 on the basis of log file analysis, after the experiment presented in Table 2.

Each video for analyzing and tuning the novelty detection algorithm contains one or two persons showing and covering their faces. The first time in the case when the face is novel, the novelty score is large (most often equal to 1), whereas the second and following times (after the training process), the novelty score is small. These values were used to establish the *T_n_* threshold.

For the novelty detection algorithm’s proper work, the initial model should contain some pretrained classes trained classically to achieve model outputs that allow proper novelty scores (similar to the novelty detection model). Therefore, the initial model pretrained with 105 celebrities (the same as in the preliminary research) was used in our experiments.

## 5. Testing of the Created System on Video with Realistic Scenes

An optional testing code was introduced into the created system, which allowed the saving of one representative face image of each track when the track recognized the face or a new class was assigned based on the image of the face.

A video of Polish news [32] and a video of the Polish presidential debate recorded in 2020 [33] were downloaded from the Internet and used as input to the system. These films were chosen because of multiple persons appearing several times, with their faces directed to the camera (in the second case), which is good material for novelty detection algorithm testing, one of the key elements of the created system. The tracking agent with recognized or assigned data to a given class is represented by one face image, saved into a folder with a name representing the given class (for instance, person105) and a file name with an agent identifier.

Table 2 shows the results of the testing system in the video file of the Polish news [32]. The first 2000 frames of film were omitted (due to unnatural presentation in the film’s introduction). Each row in the table represents the results as either created as novel, or as a further recognized class (i.e., person105), where the face images representing tracking agents are presented (named using agent ID). For instance, in the row that represents person 105 we have 2 photos (for agent 0 and agent 2). Agent 0 has created a new class (person105), whereas agent 2 has recognized an existing class (person105). As we can see, each class only contains photos obtained from one person (therefore, there is no incorrect recognition). Unfortunately, the classes of person110 and person113 belong to the same person (PF Andrzej Duda, President of Poland). The reason for this mistake was the incorrect output of the novelty detection algorithm from the number 29 agent.. The reason of this mistake could be that images for agents 15 and 23 are significantly darker than images for agent 29. The classifications for agents 30, 32 and 38 are consequences of this mistake (after the working of agent 29, there are 2 classes that belong to 1 person). Therefore, we can conclude that 22 agents out of 23 (with an identified class or new class assigned) were working correctly (a total of 95.65%).

Table 3 shows the results of the testing system in the video file of the Polish presidential debate [33]. Each row in the table represents the results as created as novel, or as further recognized classes (e.g., person105) where face images representing tracking agents are presented (named using agent ID). For instance, we have 6 photos in the row representing person105 (for agents 0, 14, 28, 37, 54 and 58). This is because agent 0 has created a new class (person105), whereas the other mentioned agents have recognized an existing class (person105). As we can see, each class contains only photos that come from one person (therefore, there is no incorrect recognition).

Moreover, each person was assigned to one newly created class; as a result, we can conclude that the novelty detection algorithm was working properly in each situation. In almost all 69 agents, excluding agent 63, the identifier was established (by training using a new identifier or classification). All persons who appeared in the video were detected, except for the meeting chairman and the woman inserted artificially into the video (the sign language interpreter); the two undetected persons had detected faces in a resolution smaller than the threshold defined by the system.

In the case of this experiment, all the novelty data have a novelty score equal to one, and no novel data have a score of 0; therefore, in this case, all thresholds $T_{n}$ between zero and one will cause the same results of the system’s work. The significantly better novelty scores in the presidential debate (presented in Table 3) are the result of the fact that in this film, all images come from one source and were captured under the same conditions in contrast to the experiment presented in Table 2 where the images came from different sources and were captured in different conditions.

The experiments presented in Table 2 performed on film containing 18,170 frames (20,170 frames with minus 2000 frames omitted) and lasted 25 min and 37 s, which displayed an average of 11.82 frames per second. The following environment was used in the test: a processor Intel^®®^ Core^®®^ i7—9750 H 2.6 GHz i 32 GB RAM, GPU NVidia GeForce RTX 2070 (8 GB GPU memory). The experiments presented in Table 3 performed on film containing 18,170 frames and lasted 6 h, 16 min and 24 s, which displayed an average of 5.2 frames per second. The following environment was used in the test: a processor Intel(R) Xeon(R) Gold 5220 CPU @ 2.20 GHz, 94 GB RAM—DDR4-2933, GPU Tesla V100-PCIE-32 GB.

## 6. Conclusions

An autonomous face training system working in real-time was created. The system is equipped with a face tracking system, the initial model of which can recognize an initial set of faces (105 celebrities) and be updated automatically using new classes (faces which have appeared), using a new novelty detection algorithm based on the SVM model. The face recognition model consists of a pretrained ResNet50 model without dense layers and a Multinomial Naïve Bayes classifier. The novelty detection algorithm (one of the advantages of the created system) can recognize whether or not the tracked face is novel (the class does not exist in the model), based on the model’s response structure. If the class is novel, the system automatically trains the model in real-time using data from the camera. The system was tested on images obtained from the camera, as well as from video files. The final tests showed that the system could automatically recognize and update the model with a new person. The difference between this created system and other systems presented in the bibliography is the possibility of it learning autonomously from a camera or video, without previously collected data.

The system developed during the research described in this paper will be a part of a greater virtual assistant system; therefore, the new persons trained autonomously will be named properly after acquiring their names another way (e.g., asking them by voice). After person identification, this part of the system could be connected with other components; for instance, recognized emotions detected on the face could be assigned to this person. Alternatively, the system could be used in an autonomous robot that serves multiple people (e.g., a robot waiter) in a similar way as in a virtual assistant system or monitoring system.

In future works, we plan to improve the novelty detection algorithm to achieve better accuracy and implement a part of the system on the OAK1 device to use its camera and Vision Processing Unit.

## Figures and Tables

**Figure 2 sensors-23-05554-f002:**
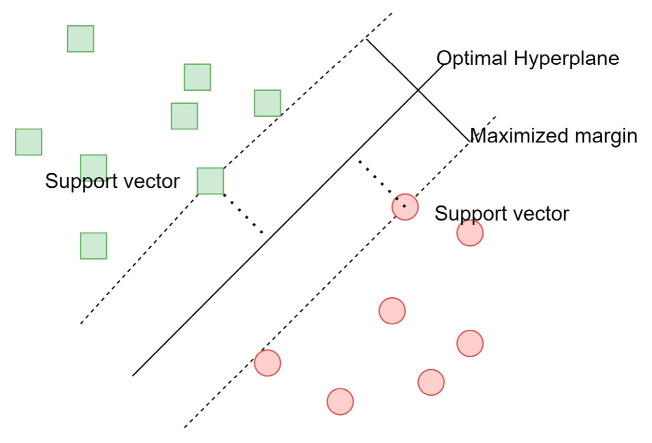
Support vector machines’ margin.

**Figure 3 sensors-23-05554-f003:**
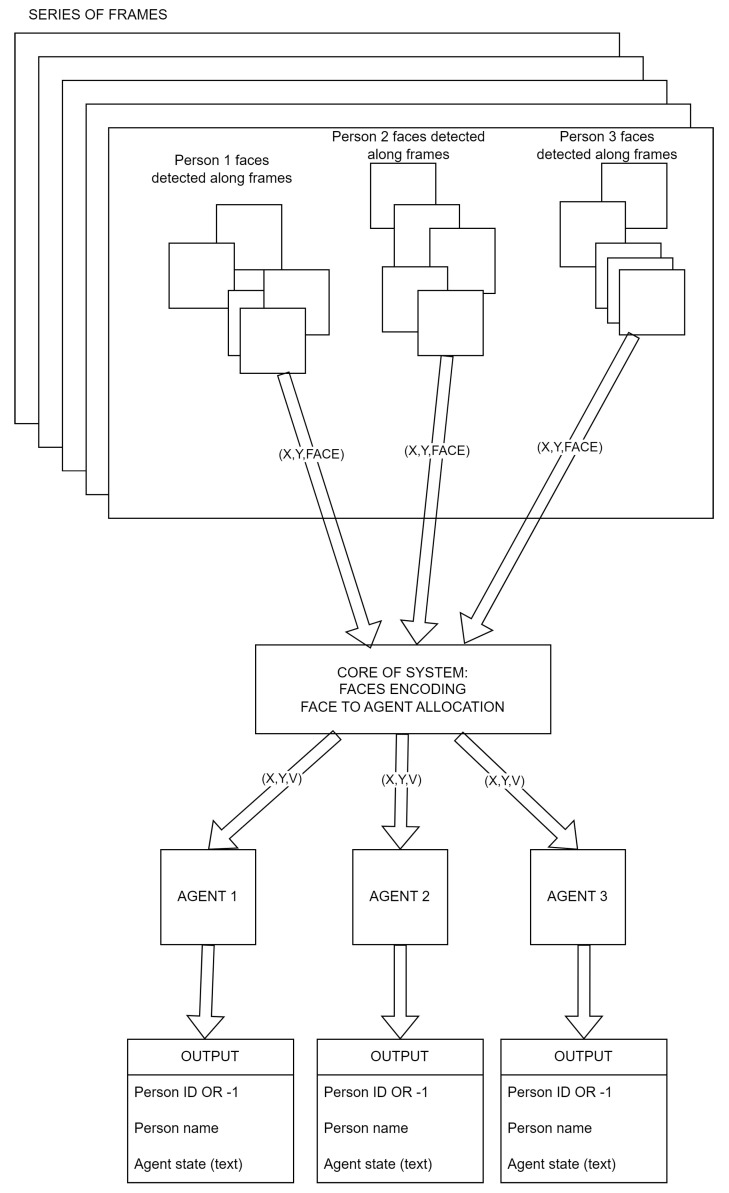
System core operation.

**Figure 4 sensors-23-05554-f004:**
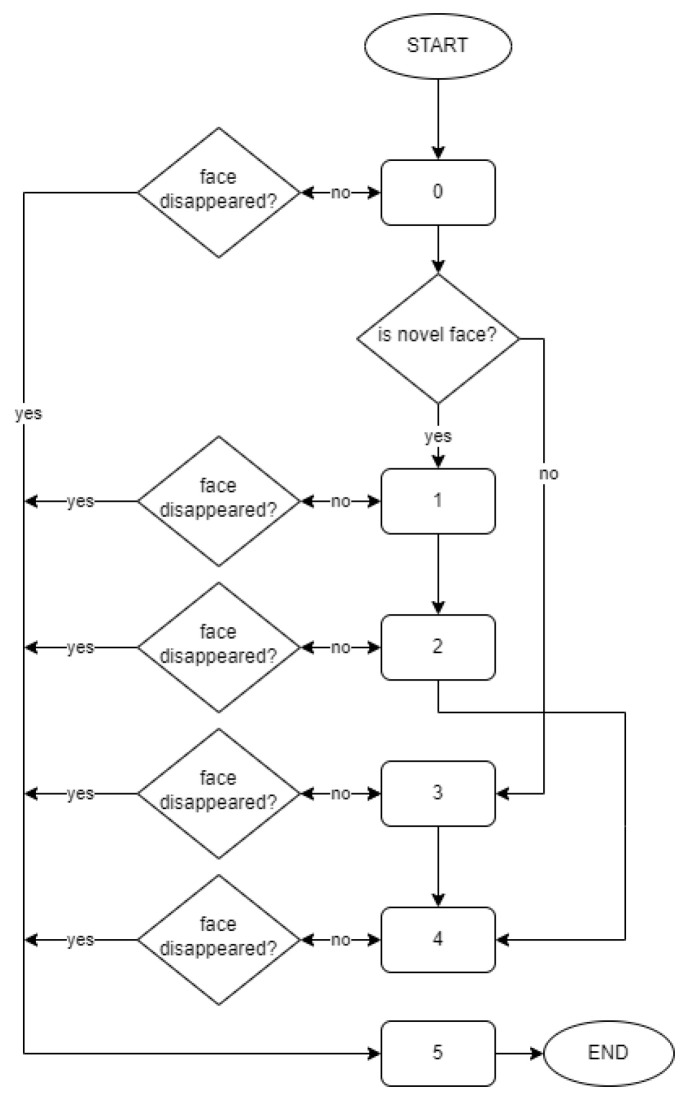
Diagram of the flow between stages in cognitive tracking agent machine state.

**Table 1 sensors-23-05554-t001:** Results (accuracy metric) of initial experiments of classification training using different classifiers on encoded faces (from a dataset of 105 celebrities).

Classifier	Test Accuracy	Update Possibility	Training 1 Sample [ms]
Multinomial Naïve Bayes	99.3761%	Yes	0.0101
Bernoulli Naïve Bayes	98.8656%	Yes	0.0278
Complement Naïve Bayes	89.5065%	Yes	0.0119
Gaussian Naïve Bayes	95.0085%	Yes	0.0089
MLP	98.1282%	Yes	0.8054
SGD	98.5820%	Yes	2.2758
SVM	99.3193%	No	1.8799
Decision tree	62.7907%	No	11.8947
Extra tree	98.2984%	No	1.4032

**Table 2 sensors-23-05554-t002:** Results of system operation based on video with news downloaded from the Internet.

Person Name	Photos Represents Tracking Agents for Identified or Created Classes
Person105	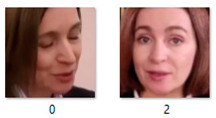 (A0, PA, 1.0), (A2, PA, 0.3)
Person106	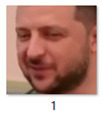 (A1, PB, 0.9)
Person107	(A3,PC,1.0), (A4,PC,0.0), (A5,PC,0.0), (A6,PC,0.8)
Person108	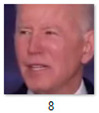 (A8,PD,1.0)
Person109	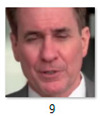 (A9,PE,1.0)
Person110	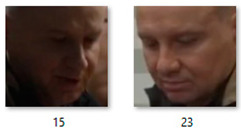 (A15,PF,1.0), (A23,PF,0.0)
Person111	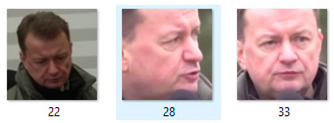 (A22,PG,1.0), (A28,PG,0.0), (A33,PG,0.0)
Person112	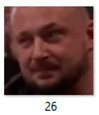 (A26,PH,1.0),
Person113	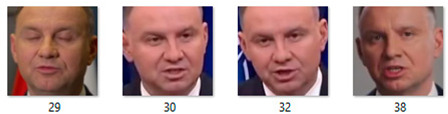 **(A29,PF,1.0)**, (A30,PF,0.0),(A32,PF,0.0),(A38,PF,0.0)
Person114	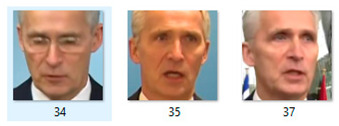 (A34,PI,1.0), (A35,PI,0.0), (A37,PI,0.0)
Person115	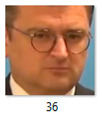 (A36,PJ,1.0)

**Table 3 sensors-23-05554-t003:** Results of system operation based on a video downloaded from the Internet.

Person Name	Photos Represent Tracking Agents for Identified or Created Classes
Person105	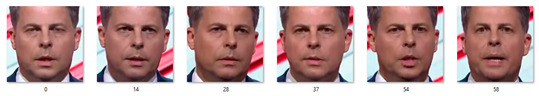 (A0,PA,1.0), (A14, PA,0.0), (A28,PA,0.0), (A37,PA,0.0), (A54,PA,0.0), (A58,PA,0.0)
Person106	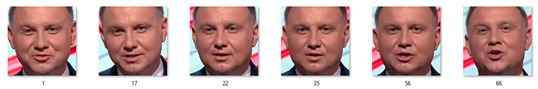 (A1,PB,1.0), (A17,PB,0.0), (A22,PB,0.0), (A35,PB,0.0), (A56,PB,0.0), (A66,PB,0.0)
Person107	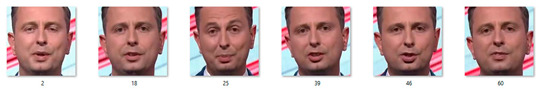 (A2,PC,1.0), (A18,PC,0.0), (A25,PC,0.0), (A39,PC,0.0), (A46,PC,0.0), (A46,PC,0.0), (A60,PC,0.0)
Person108	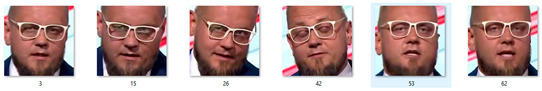 (A3,PD,1.0), (A15,PD,0.0), (A26,PD,0.0), (A42,PD,0.0), (A53,PD,0.0), (A62,PD,0.0)
Person109	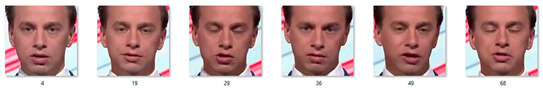 (A4,PE,1.0), (A19,PE,0.0), (A29,PE,0.0), (A36,PE,0.0), (A49,PE,0.0), (A68,PE,0.0)
Person110	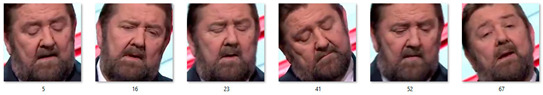 (A5,PF,1.0), (A16,PF,0.0), (A23,PF,0.0), (A41,PF,0.0), (A52,PF,0.0), (A67,PF,0.0)
Person111	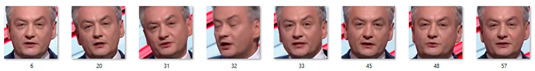 (A6,PG,1.0), (A20,PG,0.0), (A31,PG,0.0), (A32,PG,0.0), (A33,PG,0.0), (A45,PG,0.0), (A48,PG,0.0), (A57,PG,0.0)
Person112	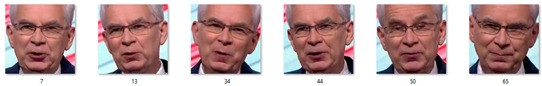 (A7,PH,1.0), (A13,PH,0.0), (A34,PH,0.0), (A44,PH,0.0), (A50,PH,0.0), (A65,PH,0.0)
Person113	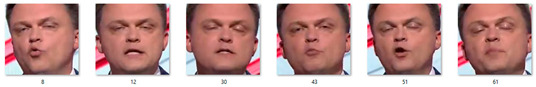 (A8,PI,1.0), (A12,PI,0.0), (A30,PI,0.0), (A43,PI,0.0), (A51,PI,0.0), (A61,PI,0.0)
Person114	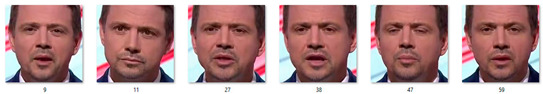 (A9,PJ,1.0), (A11,PJ,0.0), (A27,PJ,0.0), (A38,PJ,0.0), (A47,PJ,0.0), (A59,PJ,0.0)
Person115	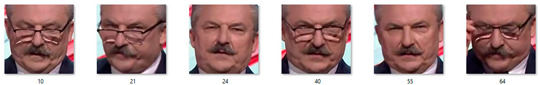 (A10,PK,1.0), (A21,PK,0.0), (A24,PK,0.0), (A40,PK,0.0), (A55,PK,0.0), (A64,PK,0.0)

## Data Availability

Available online: https://www.kaggle.com/datasets/hereisburak/pins-face-recognition (accessed on 4 May 2023) (Pins Face Recognition dataset). Available online: GitHub—WeidiXie/Keras-VGGFace2-ResNet50 (ResNet50 model pretrained weights on VGGFace2 dataset). Available online: https://www.youtube.com/watch?v=ZWZX-tDgcQE (accessed on 4 May 2023) (video file with Polish Internet News). Available online: https://www.youtube.com/watch?v=AYKGGVfCCZ0 (accessed on 4 May 2023) (video file with Polish presidential debate).

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
