# Peer review of "Autonomous Face Classification Online Self-Training System Using Pretrained ResNet50 and Multinomial Naïve Bayes"

_sensors, 2023, doi:10.3390/s23125554_

Round 1

Reviewer 1 Report

This paper presents a method for face recognition based on deep learning via ResNet50 Pretrained model.

Here are some comments: 

- The introduction or highlight of main contribution must be presented in Section 1. 

- We think the Train accuracy in table doesn't make senses. It is useless and you don't have any explanation about that. 

- A table for summarizing the dataset is welcome: resolution, different angles, number of samples of each class (person).

- The conclusion is too short and we really don't see the perspective of this work. 

- Some related and recent papers should be cited 

DOI: 10.11591/ijece.v13i4.pp4068-4075

DOI:10.1007/s00500-023-07963-x

- Different value of T_n should be tested to see its effects to the final prediction. It should be presented in a chart. 

Author Response

The responses are attached in annex

Reviewer 2 Report

The authors propose an incremental/online learning system for the problem of face recognition. In spite of the fact that the paper demonstrates a good experimental part, the presentation could and should be further improved, as follows.

Abstract

“long training process” -> relatively long training process,

 as the training speed depends on hardware characteristics

“ in a new place appears”, clarify what “new place” means

Ambiguous formulations: “one can conclude that good conditions assurance”, “The critical element of 19 system work is the novelty detection algorithm”

1.      Introduction

The considerations regarding unsupervised learning could be enhanced with references related to semi-supervised or single-shot learning.

Ambiguous formulations: could be assigned to this person in memory”

2.      Description of used methods

It refers strictly to Naïve Bayes Classifier. It should refer also to the rest of ML paradigms employed within the paper context.

3.      Initial experiments

“For instance, the model with 1000 possible classes using Scikit – learn library has a size of about 32.2MB,” – is mentioning the size important? why?

“and it is not changing while updating a model with new classes or samples from existing classes.” – which could be the explanation for that?

4.      Description of the created system

For a better understanding of the whole system, introducing a block diagram is recommended.

5.      Testing of the created system on video with realistic scenes

The system seems to work well for (quasi) frontal images. How could the system to be improved to cope with semi-frontal or profile faces.

Update the references with:

- similar approaches: follow concepts like incremental/online/zero-shot/one-shot/self-supervised/semi-supervised learning (https://doi.org/10.1016/j.patrec.2011.03.019 , https://doi.org/10.1016/j.neunet.2020.12.003, https://doi.org/10.23919/MVA.2017.7986845 ).

- tools (https://github.com/online-ml/river, https://github.com/VowpalWabbit/vowpal_wabbit)

 Eventually, create a GitHub repository for code reproducibility and testing.

included in the above comments 

Author Response

The responses are attached in the annex

Reviewer 3 Report

The proposed approach has some novelty in contribution and methodology. But, revision in terms of technical details is needed before acceptance. Also, paper organization can be improved. In this respect, some comments are suggested to describe technical details.

1. The title is not appropriate. A classification system is proposed in this paper, so it is not needed to use the phrase “training” in the title.

2. How do you select the score threshold in the Equation 8? Did you evaluate performance in terms of different threshold values?

3. I didn’t find any performance evaluating experiment in the text. Discuss about reasons in clear way. How do you evaluate the performance of your proposed approach exactly?

4. It is better to review related about face detection and facial gender classification as possible real applications of your proposed approach. For example, I find a paper titled “Gender classification in human face images for smart phone applications based on local texture information and evaluated Kullback-Leibler divergence”, which have enough relation. Cite this paper and some other related.

5. In a dataset, the number of features of all samples are same. So, the phrase "number of times that feature I appears in a sample of class y(Nyi)", is not meaningful. I think you don’t use the phrase “feature” correctly.

Moderate editing of English language is needed 

Author Response

Responses are attached in the annex

Round 2

Reviewer 1 Report

I think it's ready for publication.

Reviewer 3 Report

The revised version is better than original submission in terms of paper organization and technical details. Most of comments are considered in the revised manuscript. It can be accepted.